# ADVERSARIAL VIDEO GENERATION ON COMPLEX DATASETS

## ABSTRACT

Generative models of natural images have progressed towards high fidelity samples by the strong leveraging of scale. We attempt to carry this success to the field of video modeling by showing that large Generative Adversarial Networks trained on the complex Kinetics-600 dataset are able to produce video samples of substantially higher complexity and fidelity than previous work. Our proposed model, Dual Video Discriminator GAN (DVD-GAN), scales to longer and higher resolution videos by leveraging a computationally efficient decomposition of its discriminator. We evaluate on the related tasks of video synthesis and video prediction, and achieve new state-of-the-art Fréchet Inception Distance for prediction for Kinetics-600, as well as state-of-the-art Inception Score for synthesis on the UCF-101 dataset, alongside establishing a strong baseline for synthesis on Kinetics-600.

## 1 INTRODUCTION

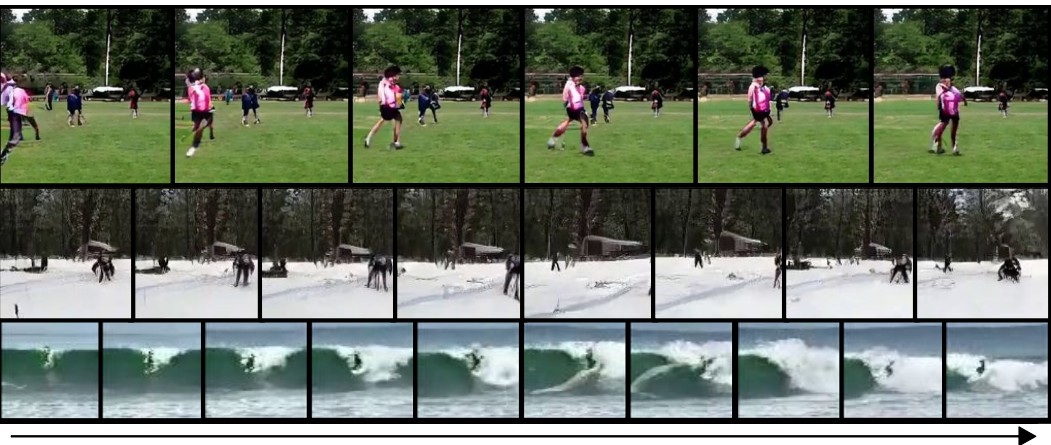

Figure 1: Selected frames from videos generated by a DVD-GAN trained on Kinetics-600 at $256 \times 256$, $128 \times 128$, and $64 \times 64$ resolutions (top to bottom).

Modern deep generative models can produce realistic natural images when trained on high-resolution and diverse datasets (Brock et al., 2019; Karras et al., 2018; Kingma & Dhariwal, 2018; Menick & Kalchbrenner, 2019; Razavi et al., 2019). Generation of natural *video* is an obvious further challenge for generative modeling, but one that is plagued by increased data complexity and computational requirements. For this reason, much prior work on video generation has revolved around relatively simple datasets, or tasks where strong temporal conditioning information is available.

We focus on the tasks of video synthesis and video prediction (defined in Section 2.1), and aim to extend the strong results of generative image models to the video domain. Building upon the state-of-the-art BigGAN architecture (Brock et al., 2019), we introduce an efficient spatio-temporal decomposition of the discriminator which allows us to train on Kinetics-600 – a complex dataset of

Figure 2: Generated video samples with interesting behavior. In raster-scan order: **a)** On-screen generated text with further lines appearing.. **b)** Zooming in on an object. **c)** Colored detail from a pen being left on paper. **d)** A generated camera change and return.

natural videos an order of magnitude larger than other commonly used datasets. The resulting model, Dual Video Discriminator GAN (DVD-GAN), is able to generate temporally coherent, high-resolution videos of relatively high fidelity (Figure 1).

Our contributions are as follows:

- We propose DVD-GAN – a scalable generative model of natural video which produces high-quality samples at resolutions up to $256 \times 256$ and lengths up to 48 frames.
- We achieve state of the art for video synthesis on UCF-101 and prediction on Kinetics-600.
- We establish class-conditional video synthesis on Kinetics-600 as a new benchmark for generative video modeling, and report DVD-GAN results as a strong baseline.

## 2 BACKGROUND

### 2.1 VIDEO SYNTHESIS AND PREDICTION

The exact formulation of the video generation task can differ in the type of conditioning signal provided. At one extreme lies unconditional video synthesis where the task is to generate any video following the training distribution. Another extreme is occupied by strongly-conditioned models, including generation conditioned on another video for content transfer (Bansal et al., 2018; Zhou et al., 2019), per-frame segmentation masks (Wang et al., 2018a), or pose information (Walker et al., 2017; Villegas et al., 2017b; Yang et al., 2018). In the middle ground there are tasks which are more structured than unconditional generation, and yet are more challenging from a modeling perspective than strongly-conditional generation (which gets a lot of information about the generated video through its input). The objective of *class-conditional video synthesis* is to generate a video of a given category (e.g., "riding a bike") while *future video prediction* is concerned with generation of continuing video given initial frames. These problems differ in several aspects, but share a common requirement of needing to generate realistic temporal dynamics, and in this work we focus on these two problems.

### 2.2 GENERATIVE ADVERSARIAL NETWORKS

Generative Adversarial Networks (GANs) (Goodfellow et al., 2014) are a class of generative models defined by a minimax game between a *Discriminator* $\mathcal{D}$ and a *Generator* $\mathcal{G}$. The original objective was proposed by Goodfellow et al. (2014), and many improvements have since been suggested, mostly targeting improved training stability (Arjovsky et al., 2017; Zhang et al., 2018; Brock et al., 2019; Gulrajani et al., 2017; Miyato et al., 2018). We use the hinge formulation of the objective (Lim & Ye, 2017; Brock et al., 2019) which is optimized by gradient descent ($\rho$ is the elementwise ReLU function):

$$\mathcal{D}: \min_{\mathcal{D}} \mathop{\mathbb{E}}_{x \sim data(x)} \Big[ \rho(1 - \mathcal{D}(x)) \Big] + \mathop{\mathbb{E}}_{z \sim p(z)} \Big[ \rho(1 + \mathcal{D}(\mathcal{G}(z))) \Big], \quad \mathcal{G}: \max_{\mathcal{G}} \mathop{\mathbb{E}}_{z \sim p(z)} \Big[ \mathcal{D}(\mathcal{G}(z)) \Big].$$

GANs have well-known limitations including a tendency towards limited diversity in generated samples (a phenomenon known as mode collapse) and the difficulty of quantitative evaluation due

to the lack of an explicit likelihood measure over the data. Despite these downsides, GANs have produced some of the highest fidelity samples across many visual domains (Karras et al., 2018; Brock et al., 2019).

### 2.3 KINETICS-600

Kinetics is a large dataset of 10-second high-resolution YouTube clips (Kay et al., 2017; DeepMind, 2018) originally created for the task of human action recognition. We use the second iteration of the dataset, Kinetics-600 (Carreira et al., 2018), which consists of 600 classes with at least 600 videos per class for a total of around 500,000 videos.[1] Kinetics videos are diverse and unconstrained, which allows us to train large models without being concerned with the overfitting that occurs on small datasets with fixed objects interacting in specified ways (Ebert et al., 2017; Blank et al., 2005). Among prior work, the closest dataset (in terms of subject and complexity) which is consistently used is UCF-101 (Soomro et al., 2012). We focus on Kinetics-600 because of its larger size (almost 50x more videos than UCF-101) and its increased diversity (600 instead of 101 classes – not to mention increased intra-class diversity). Nevertheless for comparison with prior art we train on UCF-101 and achieve a state-of-the-art Inception Score there. Kinetics contains many artifacts expected from YouTube, including cuts (as in Figure 2d), title screens and visual effects. Except when specifically described, we choose frames with stride 2 (meaning we skip every other frame). This allows us to generate videos with more complexity without incurring higher computational cost.

To the best of our knowledge we are the first to consider generative modelling of the entirety of the Kinetics video dataset[2], although a small subset of Kinetics consisting of 4,000 selected and stabilized videos (via a SIFT + RANSAC procedure) has been used in at least two prior papers (Li et al., 2018; Balaji et al., 2018). Due to the heavy pre-processing and stabilization present, as well as the sizable reduction in dataset size (two orders of magnitude) we do not consider these datasets comparable to the full Kinetics-600 dataset.

### 2.4 EVALUATION METRICS

Designing metrics for measuring the quality of generative models (GANs in particular) is an active area of research (Sajjadi et al., 2018; Barratt & Sharma, 2018). In this work we report the two most commonly used metrics, Inception Score (IS) (Salimans et al. (2016)) and Fréchet Inception Distance (FID) (Heusel et al., 2017). The standard instantiation of these metrics is intended for generative image models, and uses an Inception model (Szegedy et al., 2016) for image classification or feature extraction. For videos, we use the publicly available Inflated 3D Convnet (I3D) network trained on Kinetics-600 (Carreira & Zisserman, 2017). Our Fréchet Inception Distance is therefore very similar to the Fréchet Video Distance (FVD) (Unterthiner et al., 2018), although our implementation is different and more aligned with the original FID metric.[3] More details are in Appendix A.4.

## 3 DUAL VIDEO DISCRIMINATOR GAN

Our primary contribution is Dual Video Discriminator GAN (DVD-GAN), a generative video model of complex human actions built upon the state-of-the-art BigGAN architecture (Brock et al., 2019) while introducing scalable, video-specific generator and discriminator architectures. An overview of the DVD-GAN architecture is given in Figure 3 and a detailed description is in Appendix A.2. Unlike some of the prior work, our generator contains no explicit priors for foreground, background or motion (optical flow); instead, we rely on a high-capacity neural network to learn this in a data-driven manner. While DVD-GAN contains sequential components (RNNs), it is not autoregressive in time or in space. In other words, the pixels of each frame do not directly depend on other pixels in the video, as would be the case for auto-regressive models or models generating one frame at a time.

Generating long and high resolution videos is a heavy computational challenge: individual samples from Kinetics-600 (just 10 seconds long) contain upwards of 16 million pixels which need to be

---

[1] Kinetics is occasionally pruned and so we cannot give an exact size.

[2] In parallel with the concurrent work of Weissenborn et al. (2019).

[3] We use 'avgpool' features (rather than logits) by default, our I3D model is trained on Kinetics-600 (rather than Kinetics-400), and we pre-calculate ground-truth statistics on the entire training set.

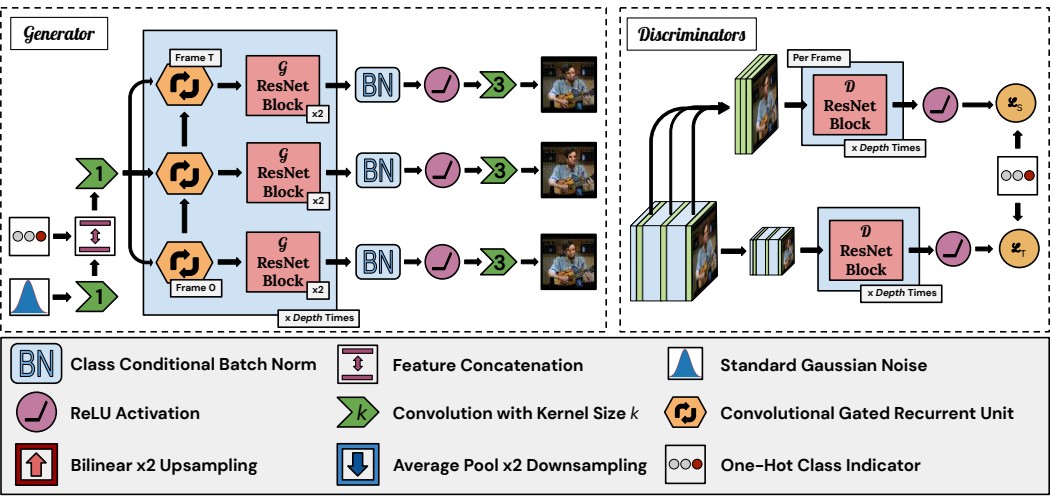

Figure 3: Simplified architecture diagram of $\mathcal{G}$ (left) and $\mathcal{D}_S/\mathcal{D}_T$ (right). More details in A.2.

generated in a consistent fashion. This is a particular challenge to the discriminator. For example, a generated video might contain an object which leaves the field of view and incorrectly returns with a different color. Here, the ability to determine this video is generated is only possible by comparing two different spatial locations across two (potentially distant) frames. Given a video with length $T$, height $H$, and width $W$, discriminators that process the entire video would have to process all $H \times W \times T$ pixels – limiting the size of the model and the size of the videos being generated.

## 3.1 DUAL DISCRIMINATORS

DVD-GAN tackles this scale problem by using two discriminators: a *Spatial Discriminator* $\mathcal{D}_S$ and a *Temporal Discriminator* $\mathcal{D}_T$. $\mathcal{D}_S$ critiques single frame content and structure by randomly sampling $k$ full-resolution frames and judging them individually. We use $k = 8$ and discuss this choice in Section 4.3. $\mathcal{D}_S$'s final score is the sum of the per-frame scores. The temporal discriminator $\mathcal{D}_T$ must provide $\mathcal{G}$ with the learning signal to generate movement (something not evaluated by $\mathcal{D}_S$). To make the model scalable, we apply a spatial downsampling function $\phi(\cdot)$ to the whole video and feed its output to $\mathcal{D}_T$. We choose $\phi$ to be $2 \times 2$ average pooling, and discuss alternatives in Section 4.3. This results in an architecture where the discriminators do not process the entire video's worth of pixels, since $\mathcal{D}_S$ processes only $k \times H \times W$ pixels and $\mathcal{D}_T$ only $T \times \frac{H}{2} \times \frac{W}{2}$. For a 48 frame video at $128 \times 128$ resolution, this reduces the number of pixels to process per video from 786432 to 327680: a 58% reduction. Despite this decomposition, the discriminator objective is still able to penalize almost all inconsistencies which would be penalized by a discriminator judging the entire video. $\mathcal{D}_T$ judges any temporal discrepancies across the entire length of the video, and $\mathcal{D}_S$ can judge any high resolution details. The only detail the DVD-GAN discriminator objective is unable to reflect is the temporal evolution of pixels within a $2 \times 2$ window. We have however not noticed this affecting the generated samples in practice. DVD-GAN's $\mathcal{D}_S$ is similar to the per-frame discriminator $\mathcal{D}_I$ in MoCoGAN (Tulyakov et al., 2018). However MoCoGAN's analog of $\mathcal{D}_T$ looks at full resolution videos, whereas $\mathcal{D}_S$ is the only source of learning signal for high-resolution details in DVD-GAN. For this reason, $\mathcal{D}_S$ is essential when $\phi$ is not the identity, unlike in MoCoGAN where the additional per-frame discriminator is less crucial.

## 3.2 RELATED WORK

Generative video modeling is a widely explored problem which includes work on VAEs (Babaeizadeh et al., 2018; Denton & Fergus, 2018; Lee et al., 2018; Hsieh et al., 2018) and recurrent models (Wang et al., 2018b; Finn et al., 2016; Wang et al., 2018c; Byeon et al., 2018), auto-regressive models (Ranzato et al., 2014; Srivastava et al., 2015; Kalchbrenner et al., 2017; Weissenborn et al., 2019), normalizing flows (Kumar et al., 2019), and GANs (Mathieu et al., 2015; Vondrick et al., 2016; Saito et al., 2017; Saito & Saito, 2018). Much prior work considers decompositions which model the

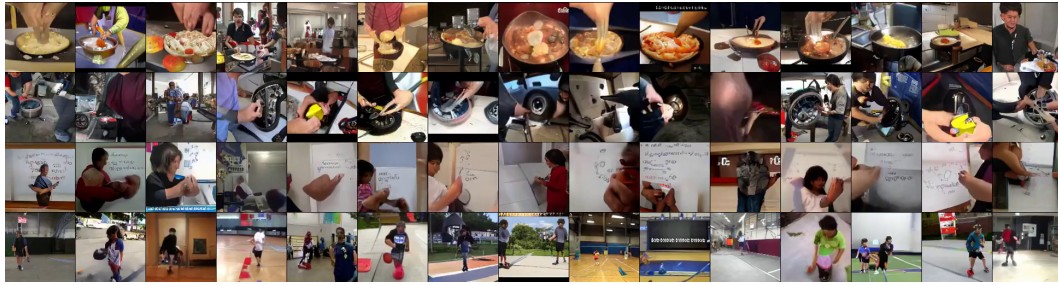

Figure 4: Each row is the first frame of 15 videos from a random class, all from the same checkpoint. The classes are: **cooking scallops**, **changing wheel (not on bike)**, **calculating**, **dribbling basketball**.

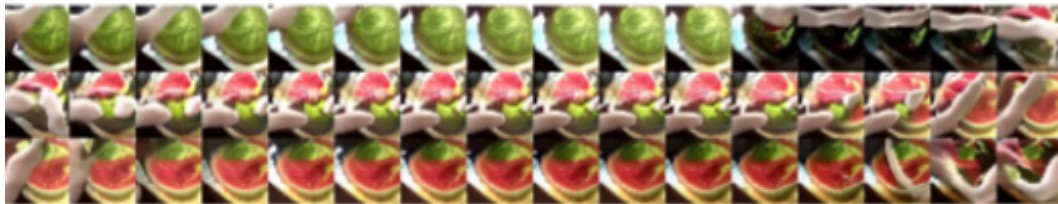

Figure 5: All 48 frames (in raster-scan order) from a $64 \times 64$ sample from watermelon cutting class.

texture and spatial consistency of objects separately from their temporal dynamics. One approach is to split $\mathcal{G}$ into foreground and background models (Vondrick et al., 2016; Spampinato et al., 2018), while another considers explicit or implicit optical flow or motion in either $\mathcal{G}$ or $\mathcal{D}$ (Saito et al., 2017; Ohnishi et al., 2018). Other methods decompose the generator (or encoder) to treat concepts like pose, content and motion separately from one another (Denton et al., 2017; Villegas et al., 2017a). Similar to DVD-GAN, MoCoGAN (Tulyakov et al., 2018) discriminates individual frames in addition to a discriminator which operates on fixed-length $K$-frame slices of the whole video (where $K < T$). Though this potentially reduces the number of pixels to discriminate to $(H \times W) + (K \times H \times W)$, Tulyakov et al. (2018) describes discriminating sliding windows, which increases the total number of pixels. Other models follow this approach by discriminating groups of frames (Xie et al., 2018; Sun et al., 2018; Balaji et al., 2018).

TGANv2 (Saito & Saito, 2018) proposes "adaptive batch reduction" for efficient training, an operation which randomly samples subsets of videos within a batch and temporal subwindows within each video. This operation is applied throughout TGANv2's $\mathcal{G}$, with heads projecting intermediate feature maps directly to pixel space before applying batch reduction, and corresponding discriminators evaluating these lower resolution intermediate outputs. An effect of this choice is that TGANv2 discriminators only evaluate full-length videos at very low resolution. We show in Figure 6 that a similar reduction in DVD-GAN's resolution when judging full videos leads to a loss in performance. We expect further reduction (towards the resolution at which TGANv2 evaluates the entire length of video) to lead to further degradation of DVD-GAN's quality. Furthermore, this method is not easily adapted towards models with large batch sizes divided across a number of accelerators, with only a small batch size per replica.

## 4 EXPERIMENTS AND ANALYSIS

A detailed description of our training setup is in Appendix A.3. Each DVD-GAN was trained on TPU pods (Google, 2018) using between 32 and 512 replicas with an Adam (Kingma & Ba, 2014) optimizer. Video Synthesis models are trained for around 300,000 learning steps, whilst Video Prediction models are trained for up to 1,000,000 steps. Most models took between 12 and 96 hours to train.

## 4.1 VIDEO SYNTHESIS

Our primary results concern the problem of *Video Synthesis*. We provide our results for the UCF-101 and Kinetics-600 datasets. With Kinetics-600 emerging as a new benchmark for generative video modelling, our results establish a strong baseline for future work.

### 4.1.1 KINETICS-600 RESULTS

Table 1: FID/IS for DVD-GAN on Kinetics-600 Video Synthesis. We present the scores of the model taken at the point in training when the best FID was attained. The "No Truncation" columns contain the scores obtained without the truncation trick. The "With Truncation" columns contain the scores obtained at the truncation level which results in the best Inception Score.

| (# Frames / Resolution) | No Truncation | | With Truncation | |
|---|---|---|---|---|
| | FID ($\downarrow$) | IS ($\uparrow$) | FID ($\downarrow$) | IS ($\uparrow$) |
| $12/64 \times 64$ | 0.85 | 53.81 | 7.13 | 187.23 |
| $12/128 \times 128$ | 1.16 | 77.45 | 13.04 | 246.18 |
| $12/256 \times 256$ | 2.05 | 62.78 | 10.17 | 162.44 |
| $48/64 \times 64$ | 13.75 | 104.09 | 47.86 | 264.12 |
| $48/128 \times 128$ | 28.44 | 81.41 | 45.79 | 188.32 |

In Table 1 we show the main result of this paper: benchmarks for Class-Conditional Video Synthesis on Kinetics-600. In this regime, we train a single DVD-GAN on all classes of Kinetics-600, supplying per-sample class information to both $\mathcal{G}$ and $\mathcal{D}$. We consider a range of resolutions and video lengths, and measure Inception Score and Fréchet Inception Distance (FID) for each (as described in Section 2.4). We further measure each model along a truncation curve, which we carry out by calculating FID and IS statistics while varying the standard deviation of the latent vectors between 0 and 1. There is no prior work with which to quantitatively compare these results (for comparative experiments see Section 4.1.2 and Section 4.2.1), but we believe these samples to show a level of fidelity not yet achieved in datasets as complex as Kinetics-600 (see samples from each row in Appendix D.1). Because all videos are resized for the I3D network (to $224 \times 224$), it is meaningful to compare metrics across equal length videos at different resolutions. Neither IS nor FID are comparable across videos of different lengths, and should be treated as separate metrics.

Generating longer and larger videos is a more challenging modeling problem, which is conveyed by the metrics (in particular, comparing 12-frame videos across $64 \times 64$, $128 \times 128$ and $256 \times 256$ resolutions). Nevertheless, DVD-GAN is able to generate plausible videos at all resolutions and with length spanning up to 4 seconds (48 frames). As can be seen in Appendix D.1, smaller videos display high quality textures, object composition and movement. At higher resolutions, generating coherent objects becomes more difficult (movement consists of a much larger number of pixels), but high-level details of the generated scenes are still extremely coherent, and textures (even complicated ones like a forest backdrop in Figure 1a) are generated well. It is further worth noting that the 48-frame models do not see more high resolution frames than the 12-frame model (due to the fixed choice of $k = 8$ described in Section 3.1), yet nevertheless learn to generate high resolution images.

### 4.1.2 VIDEO SYNTHESIS ON UCF-101

We further verify our results by testing the same model on UCF-101 (Soomro et al., 2012), a smaller dataset of 13,320 videos of human actions across 101 classes that has previously been used for video synthesis and prediction (Saito et al., 2017; Saito & Saito, 2018; Tulyakov et al., 2018). Our model produces samples with an IS of 27.38, significantly outperforming the state of the art (see Table 2 for quantitative comparison and Appendix B.1 for more details).

Table 2: IS on UCF-101 without class conditioning.

| Method | IS (↑) |
|---|---|
| VGAN (Vondrick et al., 2016) | 8.31 ± .09 |
| TGAN (Saito et al., 2017) | 11.85 ± .07 |
| MoCoGAN (Tulyakov et al., 2018) | 12.42 ± .03 |
| ProgressiveVGAN (Acharya et al., 2018) | 14.56 ± .05 |
| TGANv2 (Saito & Saito, 2018) | 24.34 ± .35 |
| **DVD-GAN (ours)** | **27.38** ± 0.53 |

Table 3: FVD on BAIR.

| Method | FVD (↓) |
|---|---|
| SVP-FP | 315.5 |
| CDNA | 296.5 |
| SV2P | 262.5 |
| SAVP | 116.4 |
| DVD-GAN-FP (ours) | 109.8 |
| Video Transformer | **94 ± 2** |

Table 4: DVD-GAN-FP's FVD scores on Video Prediction for 16 frames of Kinetics-600 without frame skipping. The final row represents a *Video Synthesis* model generating 16 frames.

| Method | Training Set FVD (↓) | Test Set FVD (↓) |
|---|---|---|
| Video Transformer (Weissenborn et al., 2019) | - | 170 ± 5 |
| **DVD-GAN-*FP*** | **68.66 ± 0.78** | **69.15 ± 1.16** |
| DVD-GAN | 32.3 ± 0.82 | 31.1 ± 0.56 |

## 4.2 FUTURE VIDEO PREDICTION

*Future Video Prediction* is the problem of generating a sequence of frames which directly follow from one (or a number) of initial conditioning frames. Both this and video synthesis require $\mathcal{G}$ to learn to produce realistic scenes and temporal dynamics, however video prediction further requires $\mathcal{G}$ to analyze the conditioning frames and discover elements in the scene which will evolve over time. In this section, we use the Fréchet Video Distance exactly as Unterthiner et al. (2018): using the logits of an I3D network trained on Kinetics-400 as features. This allows for direct comparison to prior work. Our model, DVD-GAN-FP (Frame Prediction), is slightly modified to facilitate the changed problem, and details of these changes are given in Appendix A.5.

### 4.2.1 FRAME-CONDITIONAL KINETICS

For direct comparison with concurrent work on autoregressive video models (Weissenborn et al., 2019) we consider the generation of 11 frames of Kinetics-600 at $64 \times 64$ resolution conditioned on 5 frames, where the videos for training are not taken with any frame skipping. We show results for all these cases in Table 4. Our frame-conditional model **DVD-GAN-*FP*** outperforms the prior work on frame-conditional prediction for Kinetics. The final row labeled DVD-GAN corresponds to 16-frame class-conditional Video Synthesis samples, generated without frame conditioning and without frame skipping. The FVD of this video synthesis model is notably better.

On the one hand, we hypothesize that the synthesis model has an easier generative task: it can choose to generate (relatively) simple samples for each class, rather than be forced to continue frames taken from videos which are class outliers, or contain more complicated details. On the other hand, a certain portion of the FID/FVD metric undoubtedly comes from the distribution of objects and backgrounds present in the dataset, and so it seems that the prediction model should have a handicap in the metric by being given the ground truth distribution of backgrounds and objects with which to continue videos. The synthesis model's improved performance on this task seems to indicate that the advantage of being able to select videos to generate is greater than the advantage of having a ground truth distribution of starting frames. This result is un-intuitive, as the frame conditional model has access to strictly more information about the data distribution it is trying to recover compared to the synthesis model (despite the fact that the two models are being trained by an identical objective). This experiment favors the synthesis model for FVD, but we highlight that other models or other metrics might produce the opposite ordering.

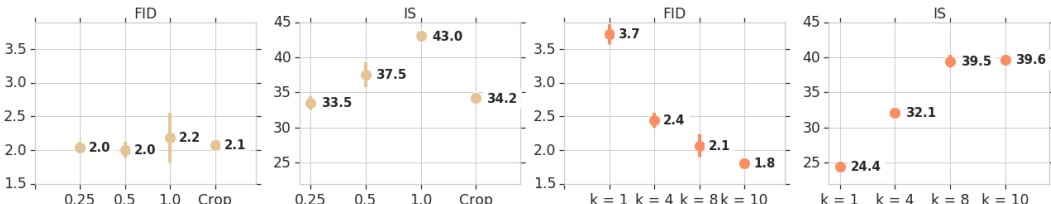

Figure 6: The effect of $\phi$ in $\mathcal{D}_T$ (left two) and $k$ in $\mathcal{D}_S$ (right two). FID is similar for any choice of $\phi$, while IS declines as downsampling increases. Increasing $k$ improves both with diminishing returns.

### 4.2.2 BAIR ROBOT PUSHING

We further test future video prediction on the single-class BAIR Robot Pushing Dataset (Ebert et al., 2017), a dataset of stationary videos of a robot arm moving around a set of changing objects. In order for direct comparison with previous results reported in Unterthiner et al. (2018), we consider generating 15 frames conditioned on a single starting frame. Like on prediction with Kinetics, we report FVD exactly as in Unterthiner et al. (2018), with ground truth statistics and conditioning frames taken from the 256-video dev set. Results are reported in Table 3. Scores are taken from Unterthiner et al. (2018). DVD-GAN-FP outperforms all prior adversarial models trained on this dataset, but performs slightly worse than Video Transformer, a concurrently developed autoregressive model Weissenborn et al. (2019). Samples from DVD-GAN-FP on BAIR are given in Figure 9.

### 4.3 DUAL DISCRIMINATOR INPUT

We analyze several choices for $k$ (the number of frames per sample in the input to $\mathcal{D}_S$) and $\phi$ (the downsampling function for $\mathcal{D}_T$). We expect setting $\phi$ to the identity or $k = T$ to result in the best model, but we are interested in the maximally compressive $k$ and $\phi$ that reduce discriminator input size (and the amount of computation), while still producing a high quality generator. For $\phi$, we consider: $2 \times 2$ and $4 \times 4$ average pooling, the identity (no downsampling), as well as a $\phi$ which takes a random half-sized crop of the input video (as in Saito & Saito (2018)). Results can be seen in Figure 6. For each ablation, we train three identical DVD-GANs with different random initializations on 12-frame clips of Kinetics-600 at $64 \times 64$ resolution for 100,000 steps. We report mean and standard deviation (via the error bars) across each group for the whole training period. For $k$, we consider 1, 2, 8 and 10 frames. We see diminishing effect as $k$ increases, so settle on $k = 8$. We note the substantially reduced IS of $4 \times 4$ downsampling as opposed to $2 \times 2$, and further note that taking half-sized crops (which results in the same number of pixels input to $\mathcal{D}_T$ as $2 \times 2$ pooling) is also notably worse.

## 5 CONCLUSION

We approached the challenging problem of modeling natural video by introducing a GAN capable of capturing the complexity of a large video dataset. We showed that on UCF-101 and frame-conditional Kinetics-600 it quantitatively achieves the new state of the art, alongside qualitatively producing video synthesis samples with high complexity and diversity. We further wish to emphasize the benefit of training generative models on large and complex video datasets, such as Kinetics-600, and envisage the strong baselines we established on this dataset with DVD-GAN will be used as a reference point by the generative modeling community moving forward. While much remains to be done before realistic videos can be consistently generated in an unconstrained setting, we believe DVD-GAN is a step in that direction.

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

Figure 7: The residual blocks for $\mathcal{G}$ and $\mathcal{D}_S/\mathcal{D}_T$. See Figure 3 for the icons and A.2 for more detail.

Yunbo Wang, Lu Jiang, Ming-Hsuan Yang, Li-Jia Li, Mingsheng Long, and Li Fei-Fei. Eidetic 3d lstm: A model for video prediction and beyond. 2018c.

Dirk Weissenborn, Oscar Täckström, and Jakob Uszkoreit. Scaling autoregressive video models. *arXiv:1906.02634*, 2019.

Yuxin Wu and Kaiming He. Group normalization. In *ECCV*, 2018.

You Xie, Erik Franz, Mengyu Chu, and Nils Thuerey. tempoGAN: A temporally coherent, volumetric GAN for super-resolution fluid flow. *TOG*, 2018.

Ceyuan Yang, Zhe Wang, Xinge Zhu, Chen Huang, Jianping Shi, and Dahua Lin. Pose guided human video generation. In *ECCV*, 2018.

Han Zhang, Ian Goodfellow, Dimitris Metaxas, and Augustus Odena. Self-attention generative adversarial networks. *arXiv:1805.08318*, 2018.

Yipin Zhou, Zhaowen Wang, Chen Fang, Trung Bui, and Tamara L Berg. Dance dance generation: Motion transfer for internet videos. *arXiv:1904.00129*, 2019.

## A  EXPERIMENT METHODOLOGY

### A.1  DATASET PROCESSING

For all datasets we randomly shuffle the training set for each model replica independently. Experiments on the BAIR Robot Pushing dataset are conducted in the native resolution of $64 \times 64$, where for UCF-101 we operate at a (downsampled) $128 \times 128$ resolution. This is done by a bilinear resize such that the video's smallest dimension is mapped to 128 pixels while maintaining aspect ratio (144 for UCF-101). From this we take a random 128-pixel crop along the other dimension. We use the same procedure to construct datasets of different resolutions for Kinetics-600. All three datasets contain videos with more frames than we generate, so we take a random sequence of consecutive frames from the resized output. For UCF-101, we augmented the dataset by randomly performing left-right flips with probability 0.5.

### A.2  ARCHITECTURE DESCRIPTION

Our model adopts many architectural choices from Brock et al. (2019) including our nomenclature for describing network width, which is determined by the product of a channel multiplier $ch$ with a constant for each layer in the network. The layer-wise constants for $\mathcal{G}$ are $[8, 8, 8, 4, 2]$ for $64 \times 64$ videos and $[8, 8, 8, 4, 2, 1]$ for $128 \times 128$. The width of the $i$-th layer is given by the product of $ch$ and the $i$-th constant and all layers prior to the residual network in $\mathcal{G}$ use the initial layer's multiplier and we refer to the product of that and $ch$ as $ch_0$. $ch$ in DVD-GAN is 128 for videos with $64 \times 64$ resolution and 96 otherwise. The corresponding $ch$ lists for both $\mathcal{D}_T$ and $\mathcal{D}_S$ are $[2, 4, 8, 16, 16]$ for $64 \times 64$ resolution and $[1, 2, 4, 8, 16, 16]$ for $128 \times 128$.

The input to $\mathcal{G}$ consists of a Gaussian latent noise $z \sim \mathcal{N}(0, I)$ and a learned linear embedding $e(y)$ of the desired class $y$. Both inputs are 120-dimensional vectors. $\mathcal{G}$ starts by computing an affine transformation of $[z; e(y)]$ to a $[4, 4, ch_0]$-shaped tensor (in Figure 3 this is represented as a $1 \times 1$ convolution). $[z; e(y)]$ is used as the input to all class-conditional Batch Normalization layers throughout $\mathcal{G}$ (the gray line in Figure 7).

This is then treated as the input (at each frame we would like to generate) to a Convolutional Gated Recurrent Unit (Ballas et al., 2015; Sutskever et al., 2011) whose update rule for input $x_t$ and previous output $h_{t-1}$ is given by the following:

$$r = \sigma(W_r \star_3 [h_{t-1}; x_t] + b_r)$$
$$u = \sigma(W_u \star_3 [h_{t-1}; x_t] + b_u)$$
$$c = \rho(W_c \star_3 [x_t; r \odot h_{t-1}] + b_c)$$
$$h_t = u \odot h_{t-1} + (1 - u) \odot c$$

In these equations $\sigma$ and $\rho$ are the elementwise sigmoid and ReLU functions respectively, the $\star_n$ operator represents a convolution with a kernel of size $n \times n$, and the $\odot$ operator is an elementwise multiplication. Brackets are used to represent a feature concatenation. This RNN is unrolled once per frame. The output of this RNN is processed by two residual blocks (whose architecture is given by Figure 7). The time dimension is combined with the batch dimension here, so each frame proceeds through the blocks independently. The output of these blocks has width and height dimensions which are doubled (we skip upsampling in the first block). This is repeated a number of times, with the output of one RNN + residual group fed as the input to the next group, until the output tensors have the desired spatial dimensions. We do not reduce over the time dimension when calculating Batch Normalization statistics. This prevents the network from utilizing the Batch Normalization layers to pass information between timesteps.

The spatial discriminator $\mathcal{D}_S$ functions almost identically to BigGAN's discriminator, though an overview of the residual blocks is given in Figure 7 for completeness. A score is calculated for each of the uniformly sampled $k$ frames (we default to $k = 8$) and the $\mathcal{D}_S$ output is the sum over per-frame scores. The temporal discriminator $\mathcal{D}_T$ has a similar architecture, but pre-processes the real or generated video with a $2 \times 2$ average-pooling downsampling function $\phi$. Furthermore, the first two residual blocks of $\mathcal{D}_T$ are 3-D, where every convolution is replaced with a 3-D convolution with a kernel size of $3 \times 3 \times 3$. The rest of the architecture follows BigGAN (Brock et al., 2019).

### A.3 TRAINING DETAILS

Sampling from DVD-GAN is very efficient, as the core of the generator architecture is a feed-forward convolutional network: two $64 \times 64$ 48-frame videos can be sampled in less than 150ms on a single TPU core. The dual discriminator $\mathcal{D}$ is updated twice for every update of $\mathcal{G}$ (Heusel et al., 2017) and we use Spectral Normalization (Zhang et al., 2018) for all weight layers (approximated by the first singular value) and orthogonal initialization of weights (Saxe et al., 2013). Sampling is carried out using the exponential moving average of $\mathcal{G}$'s weights, which is accumulated with decay $\gamma = 0.9999$ starting after 20,000 training steps. The model is optimized using Adam (Kingma & Ba, 2014) with batch size $512$ and a learning rate of $1 \cdot 10^{-4}$ and $5 \cdot 10^{-4}$ for $\mathcal{G}$ and $\mathcal{D}$ respectively. Class conditioning in $\mathcal{D}$ (Miyato & Koyama, 2018) is projection-based whereas $\mathcal{G}$ relies on class-conditional Batch Normalization (Ioffe & Szegedy, 2015; De Vries et al., 2017; Dumoulin et al., 2017): equivalent to standard Batch Normalization without a learned scale and offset, followed by an elementwise affine transformation where each parameter is a function of the noise vector and class conditioning.

### A.4 FID FOR KINETICS-600 SYNTHESIS

The FID we use for Synthesis on Kinetics-600 is calculated exactly as Fréchet Video Distance (Unterthiner et al., 2018) except that we use a different feature network: an I3D trained on Kinetics-600 (as opposed to the network trained on Kinetics-400 in FVD) and features from the final hidden layer instead of the logits. This metric can be implemented as a small change from the publically available FVD code (Google, 2019) by changing the name of the TF-Hub module to *'https://tfhub.dev/deepmind/i3d-kinetics-600/1'* and loading the tensor named *'RGB/inception_i3d/Logits/AvgPool3D'* from the resulting graph.

### A.5 ARCHITECTURE EXTENSION TO VIDEO PREDICTION

In order to provide results on future video prediction problems we describe a simple modification to DVD-GAN to facilitate the added conditioning. A diagram of the extended model is in Figure 8.

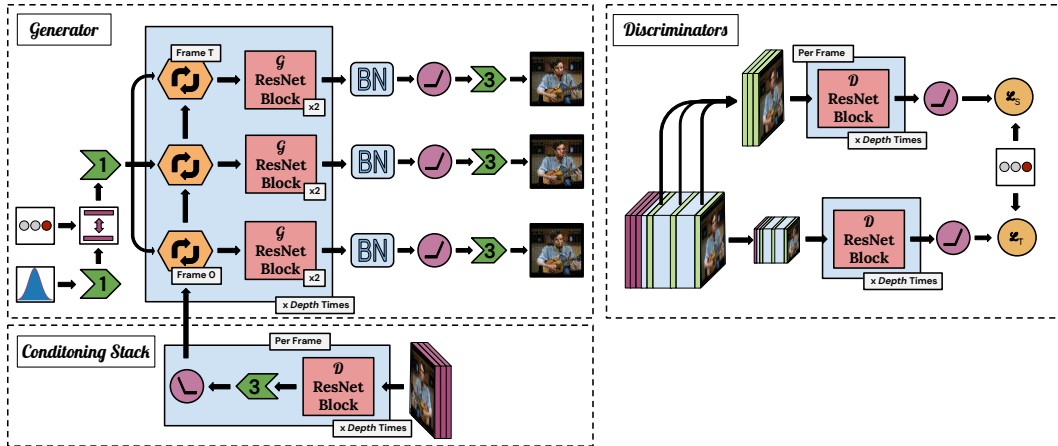

Figure 8: An architecture diagram describing the changes for the frame conditional model.

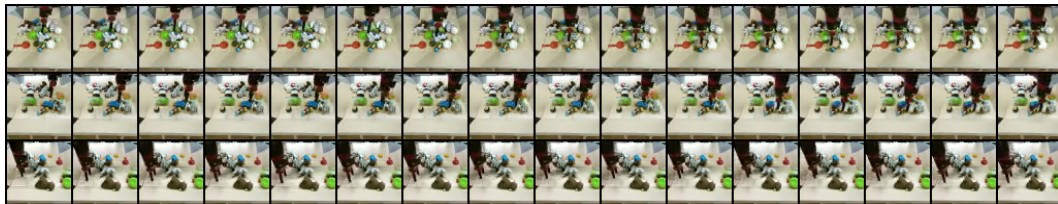

Figure 9: Three video samples from a prediction model trained on the BAIR robot pushing dataset. Each row is a separate video, the leftmost column is a (true) conditioning frame.

Given $C$ conditioning frames, our modified DVD-GAN-*FP* passes each frame separately through a deep residual network identical to $\mathcal{D}_S$. The (near) symmetric design of $\mathcal{G}$ and $\mathcal{D}_S$'s residual blocks mean that each output from a $\mathcal{D}$-style residual block has a corresponding intermediate tensor in $\mathcal{G}$ of the same spatial resolution. After each block the resulting features for each conditioning frame are stacked in the channel dimension and passed through a $3 \times 3$ convolution and ReLU activation. The resulting tensor is used as the initial state for the Convolutional GRU in the corresponding block in $\mathcal{G}$. Note that the frame conditioning stack reduces spatial resolution while $\mathcal{G}$ increases resolution. Therefore the smallest features of the conditioning frames (which have been through the most layers) are input earliest in $\mathcal{G}$ and the larger features (which have been through less processing) are input to $\mathcal{G}$ towards the end. $\mathcal{D}_T$ operates on the concatenation of the conditioning frames and the output of $\mathcal{G}$, meaning that it does not receive any extra information detailing that the first $C$ frames are special. However to reduce wasted computation we do not sample the first $C$ frames for $\mathcal{D}_S$ on real or generated data. This technically means that $D_S$ will never see the first few frames from real videos at full resolution, but this was not an issue in our experiments. Finally, our video prediction variant does not condition on any class information, allowing us to directly compare with prior art. This is achieved by settling the class id of all samples to 0.

# B  FURTHER EXPERIMENTS

## B.1  UCF-101

UCF-101 (Soomro et al., 2012) is a dataset of 13,320 videos of human actions across 101 classes that has previously been used for video synthesis and prediction (Saito et al., 2017; Saito & Saito, 2018; Tulyakov et al., 2018). In this case, DVD-GAN is not conditioned on class labels to make our results comparable with prior work. This is achieved by setting the class labels of all input samples to 0. We report Inception Score (IS) calculated with a C3D network (Tran et al., 2015)

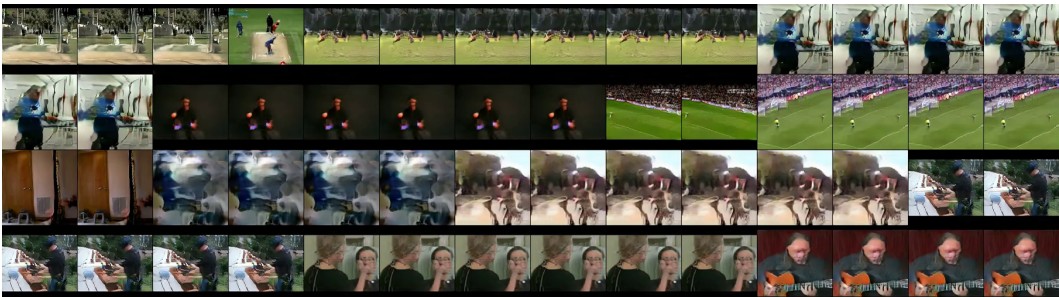

Figure 10: The first frames of interpolations between UCF-101 samples. Each row is a separate interpolation. Contrast with samples in Appendix D.2.

for quantitative comparison with prior work.[4] This evaluation is performed by re-scaling the video to $128 \times 128$, normalizing the input features based on mean statistics of the ground truth dataset, then taking a $112 \times 112$ center crop and applying C3D. Our model produces samples with an IS of $27.38$, significantly outperforming the state of the art (see Table 2). The DVD-GAN architecture on UCF-101 is identical to the model used for Kinetics, and is trained on 16-frame $128 \times 128$ clips from UCF-101.

The lack of class information does hurt the performance of DVD-GAN, and training on UCF-101 with class labels leads to an improved model with an Inception Score of $32.97$. This is directly comparable to Conditional TGAN Saito et al. (2017) which achieved an IS of $15.83$ and is close to the IS reported for the ground truth data ($34.49$). However we note than many more recent video generation papers do not test in this regime. It is worth mentioning that our improved score is, at least partially, due to memorization of the training data. In Figure 10 we show interpolation samples from our best UCF-101 model. Like interpolations in Appendix D.2, we sample 2 latents (left and rightmost columns) and show samples from the linear interpolation in latent space along each row. Here we show 4 such interpolations (the first frame from each video). Unlike Kinetics-600 interpolations, which smoothly transition from one sample to the other, we see abrupt jumps in the latent space between highly distinct samples, and little intra-video diversity between samples in each group. It can be further seen that some generated samples highly correlate with samples from the training set.

We show this both as a failure of the Inception Score metric, the commonly reported value for class-conditional video synthesis on UCF-101, but also as strong signal that UCF-101 is not a complex or diverse enough dataset to facilitate interesting video generation. Each class is relatively small, and reuse of clips from shared underlying videos means that the intra-class diversity can be restricted to just a handful of videos per class. This suggests the need for larger, more diverse and challenging datasets for generative video modelling, and we believe that Kinetics-600 provides a better benchmark for this task.

## C    MISCELLANEOUS EXPERIMENTS

Here we detail a number of modifications or miscellaneous results we experimented with which did not produce a conclusive result.

- We experimented with several variations of normalization which do not require calculating statistics over a batch of data. Group Normalization (Wu & He, 2018) performed best, almost on a par with (but worse than) Batch Normalization. We further tried Layer Normalization (Lei Ba et al., 2016), Instance Normalization (Ulyanov et al., 2016), and no normalization, but found that these significantly underperformed Batch Normalization.
- We found that removing the final Batch Normalization in $\mathcal{G}$, which occurs after the ResNet and before the final convolution, caused a catastrophic failure in learning. Interestingly, just

---

[4]We use the Chainer (Tokui et al., 2015) implementation of Inception Score for C3D available at https://github.com/pfnet-research/tgan.

removing the Batch Normalization layers within $\mathcal{G}$'s residual blocks still led to good (though slightly worse) generative models. In particular, variants without Batch Normalization in the residual blocks often achieve significantly higher IS (up to 110.05 for $64 \times 64$ 12 frame samples – twice normal). But these models had substantially worse FID scores (1.22 for the aforementioned model) – and produced qualitatively worse video samples.

- Early variants of DVD-GAN contained Batch Normalization which normalized over all frames of all batch elements. This gave $\mathcal{G}$ an extra channel to convey information across time. It took advantage of this, with the result being a model which required batch statistics in order to produce good samples. We found that the version which normalizes over timesteps independently worked just as well and without the dependence on statistics.

- Models based on the residual blocks of BigGAN-deep trained faster (in wall clock time) but slower with regards to metrics, and struggled to reach the accuracy of models based on BigGAN's residual blocks.

## D  GENERATED SAMPLES

It is difficult to accurately convey complicated generated video through still frames. Where provided, we recommend readers view the generated videos themselves via the provided links. We refer to videos within these batches by row/column number where the video in the 0th row and column is in the top left corner.

### D.1 Synthesis Samples

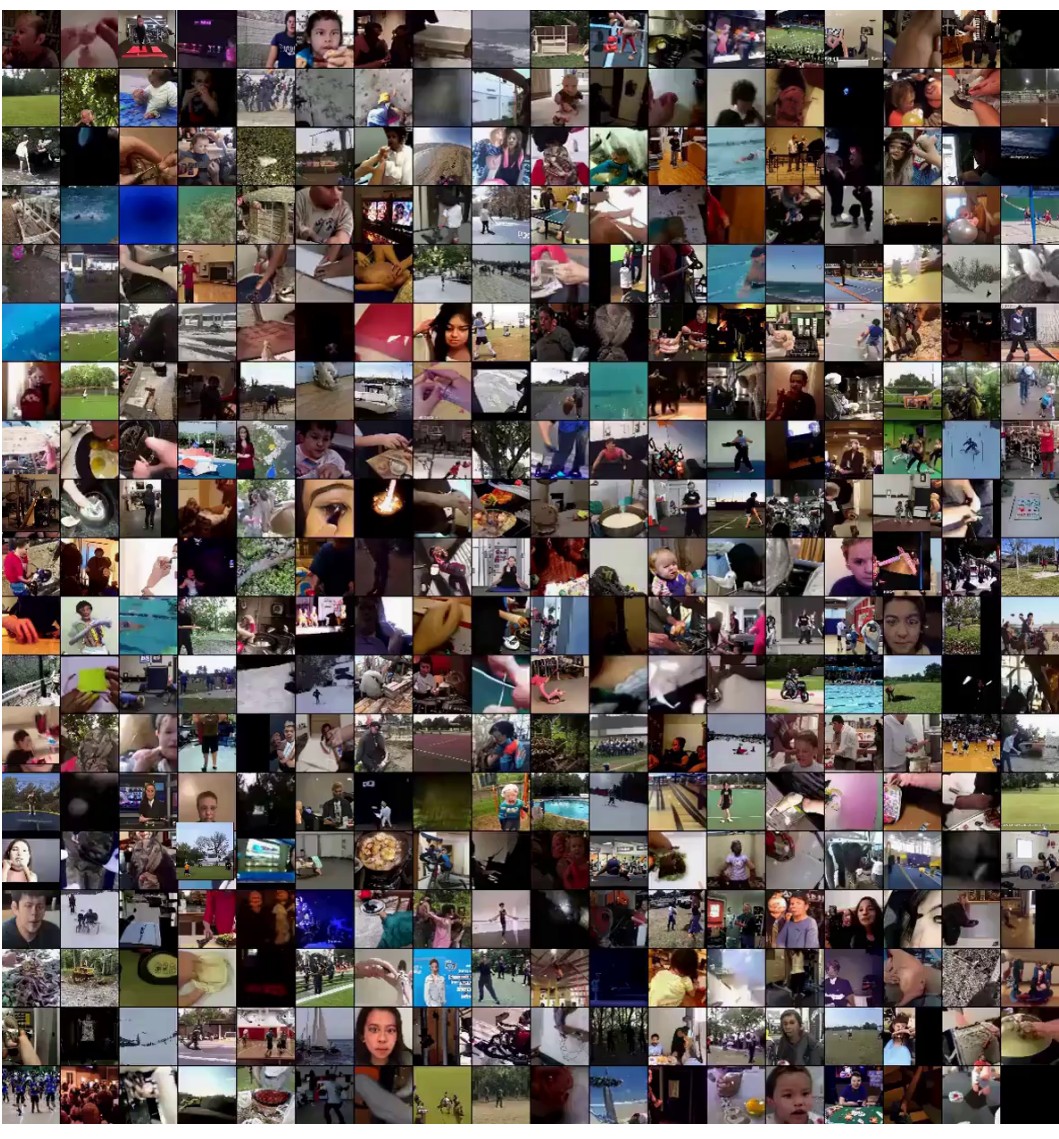

Figure 11: The first frames from a random batch of samples from DVD-GAN trained on 12 frames of $64 \times 64$ Kinetics-600. Full samples at `https://drive.google.com/file/d/155F1lkHA5fMAd7k4W3CQvTsi1eKQDhGb/view?usp=sharing`.

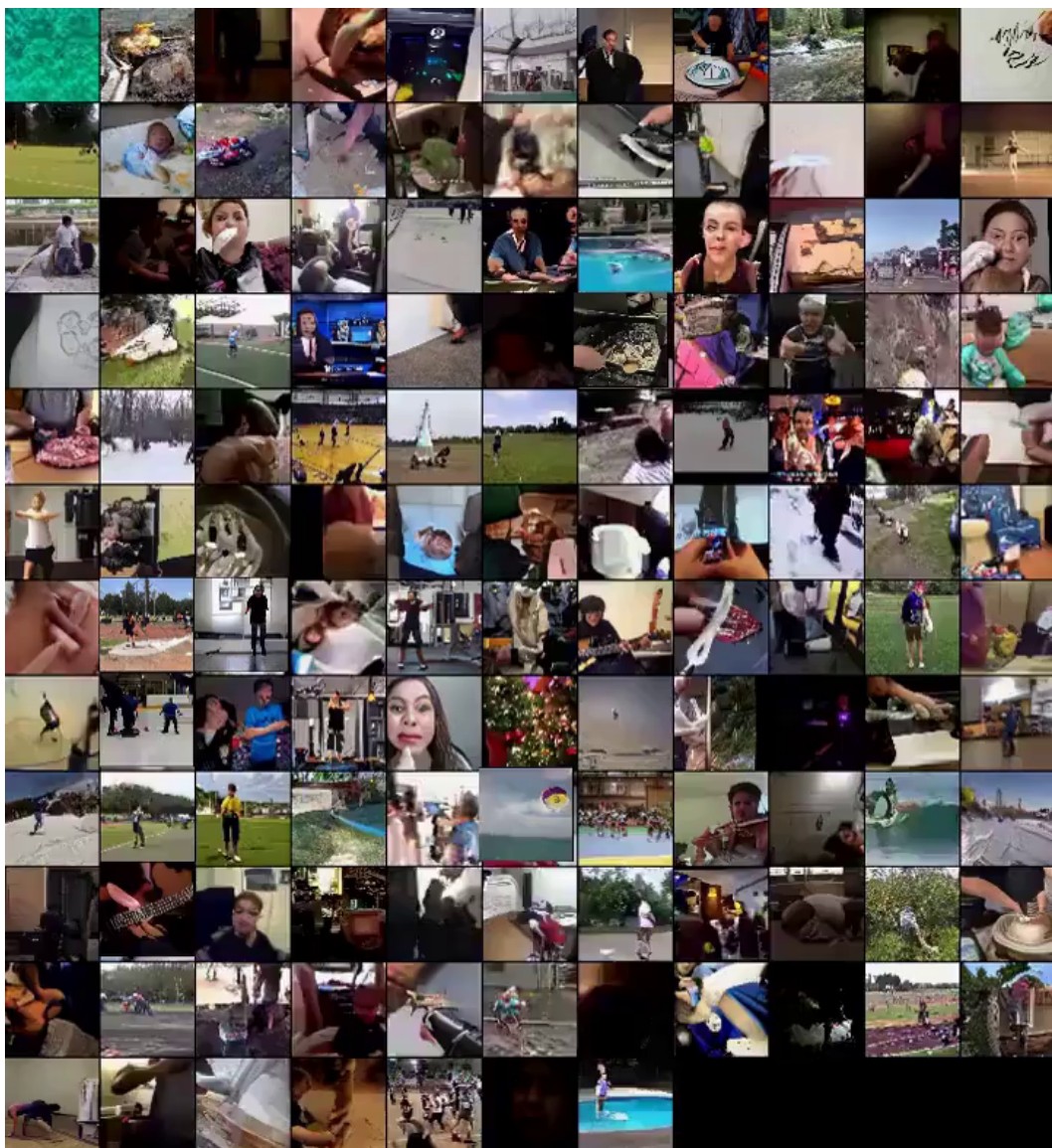

Figure 12: The first frames from a random batch of samples from DVD-GAN trained on 48 frames of $64 \times 64$ Kinetics-600. Full samples at `https://drive.google.com/file/d/1FjOQYdUuxPXvS8yeOhXdPQMapUQaklLi/view?usp=sharing`.

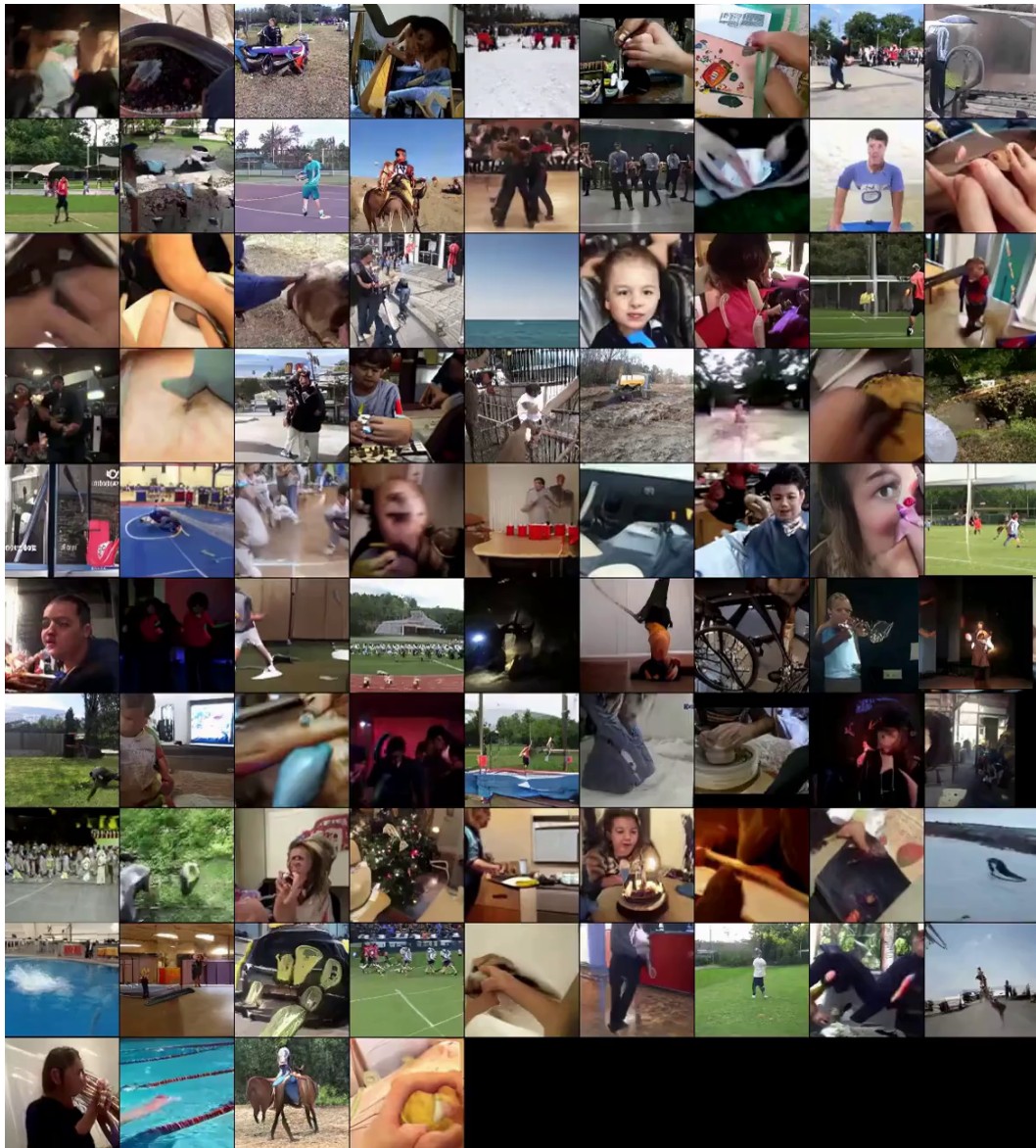

Figure 13: The first frames from a random batch of samples from DVD-GAN trained on 12 frames of $128 \times 128$ Kinetics-600. Full samples at `https://drive.google.com/file/d/165Yxuvvu3viOy-39LhhSDGtczbWphj_i/view?usp=sharing`

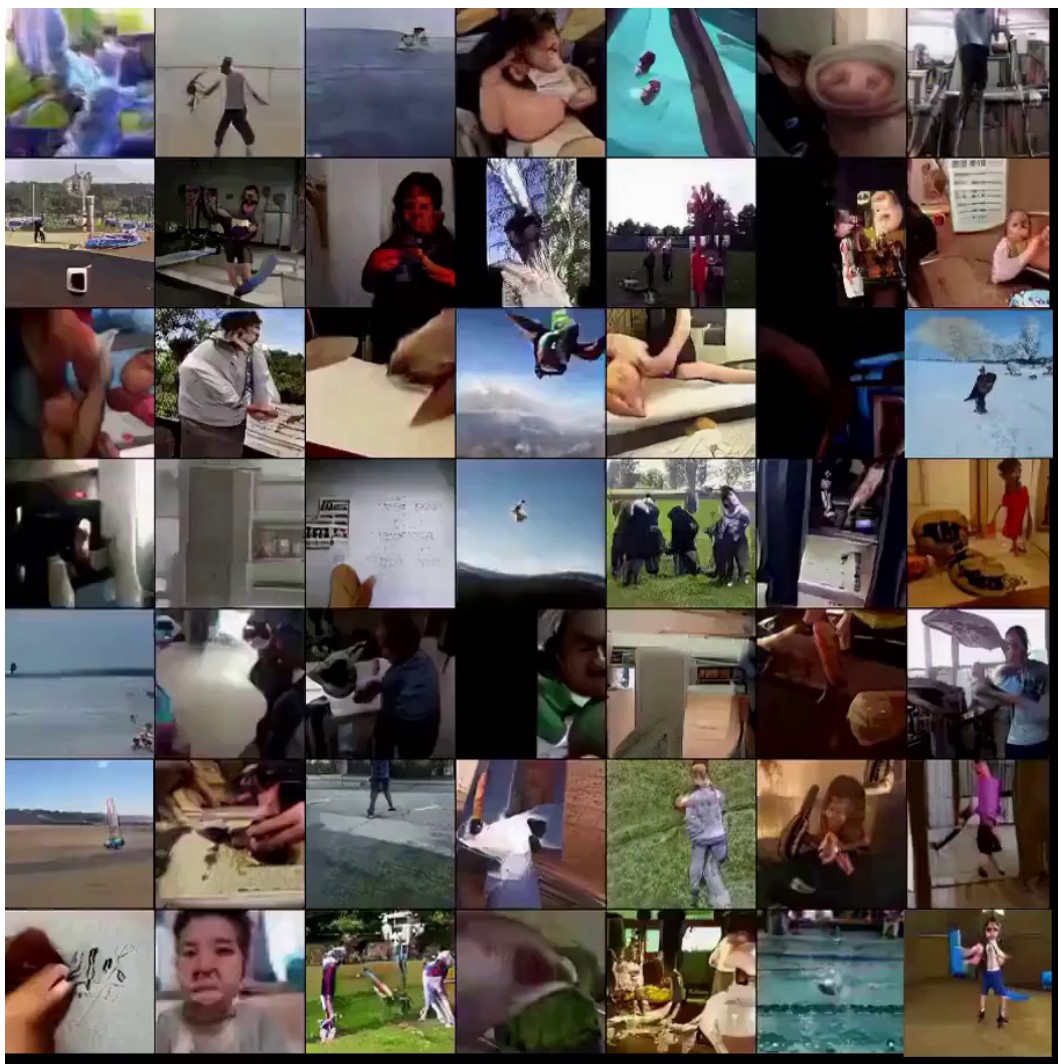

Figure 14: The first frames from a random batch of samples from DVD-GAN trained on 48 frames of $128 \times 128$ Kinetics-600. Full samples at `https://drive.google.com/file/d/1P8SsWEGP6tEGPPNPH-iVycOlN6vpIgE8/view?usp=sharing`. The sample in row 1, column 5 is a stereotypical example of a degenerate sample occasionally produced by DVD-GAN.

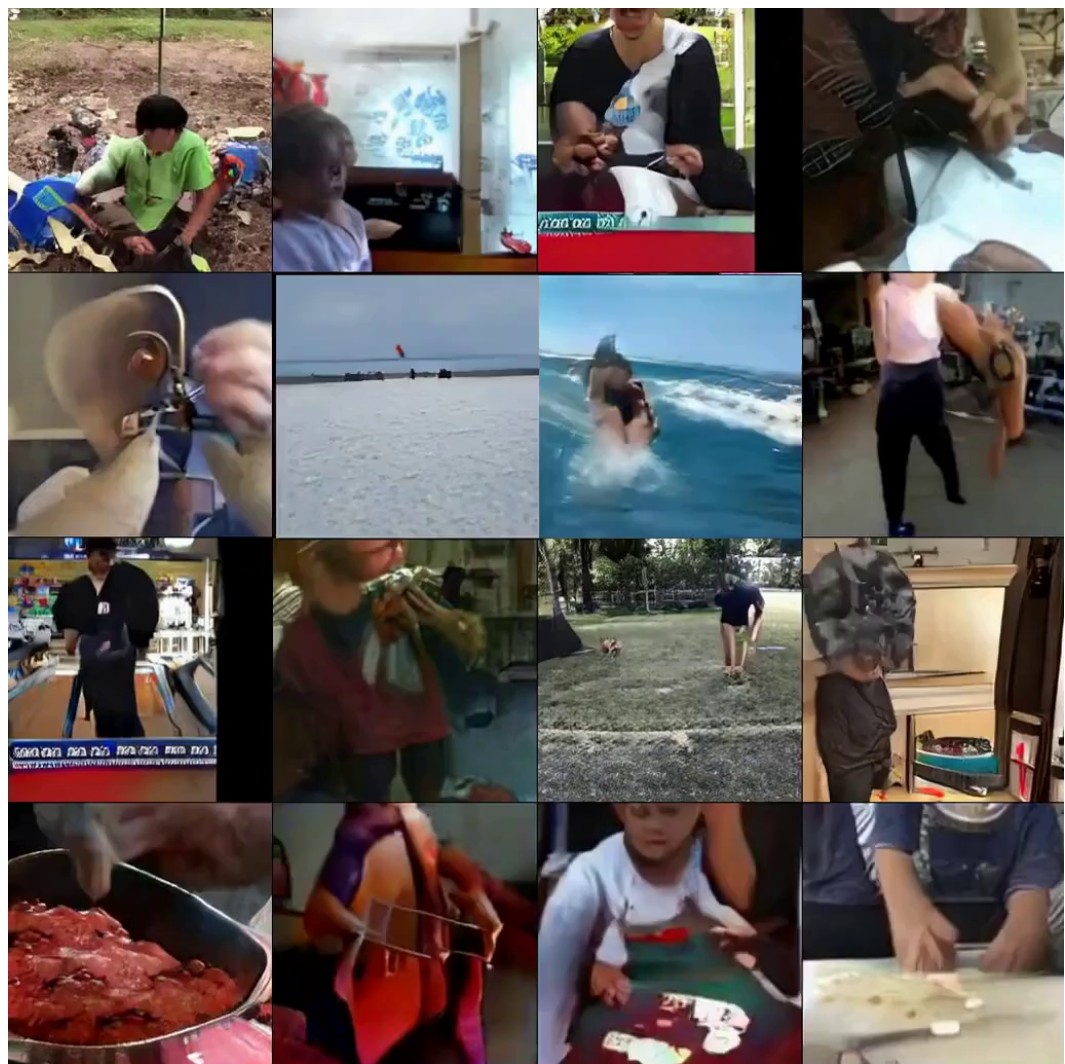

Figure 15: The first frames from a random batch of samples from DVD-GAN trained on 12 frames of $256 \times 256$ Kinetics-600. Full samples at `https://drive.google.com/file/d/1RGRVKCpVaG8z3p9GBCamRk4apiIR7jUc/view?usp=sharing`.

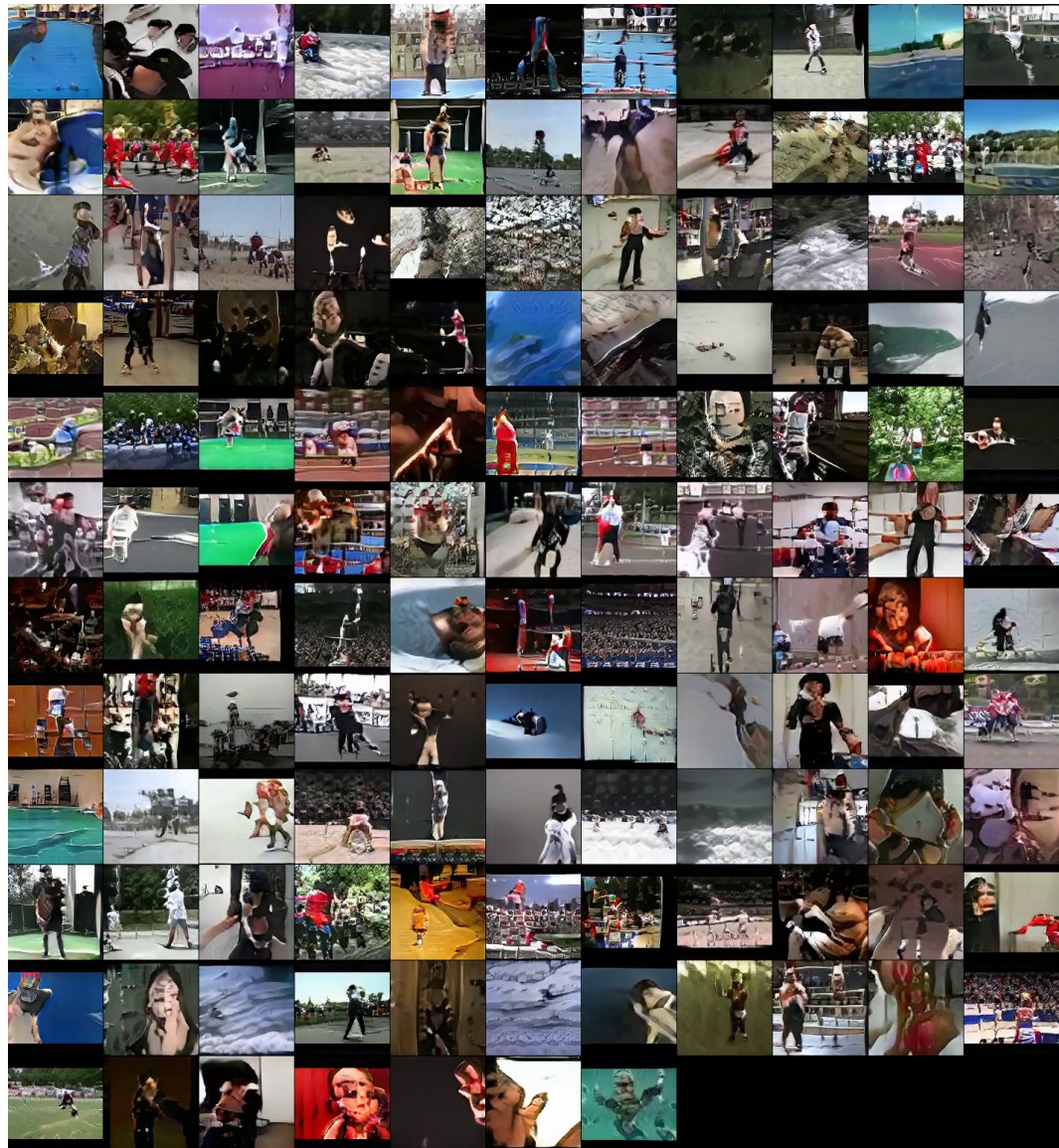

Figure 16: The first frames from a random batch of samples from DVD-GAN trained on UCF-101. Full samples at `https://drive.google.com/file/d/1VVLF3bQLyfKtIiSxaKWKq5qFRHmv5EVW/view?usp=sharing`.

## D.2 INTERPOLATION SAMPLES

We expect $\mathcal{G}$ to produce samples of higher quality from latents near the mean of the distribution (zero). This is the idea behind the Truncation Trick (Brock et al., 2019). Like BigGAN, we find that DVD-GAN is amenable to truncation. We also experiment with interpolations in the latent space and in the class embedding. In both cases, interpolations are evidence that $\mathcal{G}$ has learned a relatively smooth mapping from the latent space to real videos: this would be impossible for a network that has only memorized the training data, or which is only capable of generating a few exemplars per class. Note that while all latent vectors along an interpolation are valid (and therefore $\mathcal{G}$ should produce a reasonable sample), at no point during training is $\mathcal{G}$ asked to generate a sample halfway between two classes. Nevertheless $\mathcal{G}$ is able to interpolate between even very distinct classes.

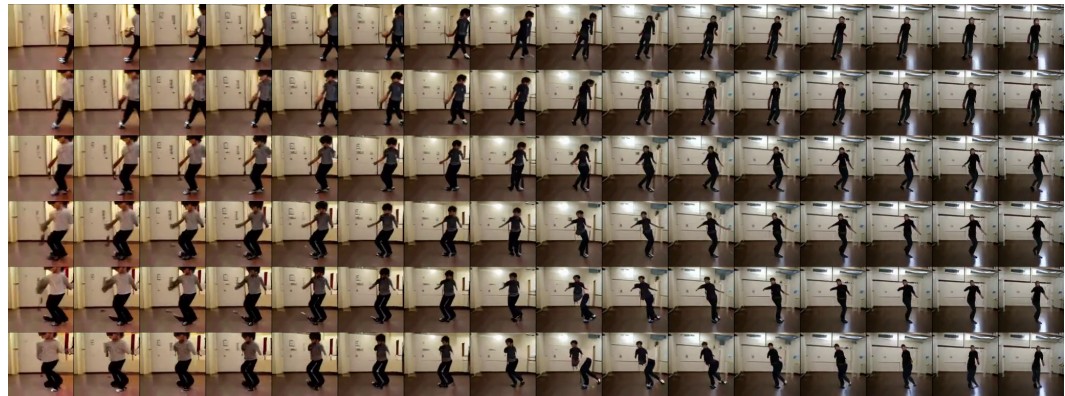

Figure 17: An example *intra-class* interpolation. Each column is a separate video (the vertical axis is the time dimension). The left and rightmost columns are randomly sampled latent vectors and are generated under a shared class. Columns in between represent videos generated under the same class across the linear interpolation between the two random samples. Note the smooth transition between videos at all six timesteps displayed here.

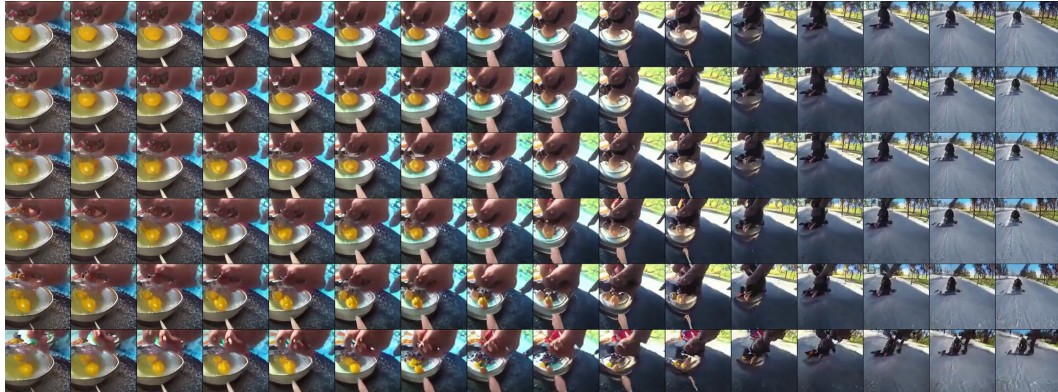

Figure 18: An example of *class* interpolation. As before, each column is a sequence of timesteps of a single video. Here, we sample a **single** latent vector, and the left and rightmost columns represent generating a video of that latent under two different classes. Columns in between represent videos of that same latent generated across an interpolation of the class embedding. Even though at no point has DVD-GAN been trained on data under an interpolated class, it nevertheless produces reasonable samples.

