# OpenReview forum: "Adversarial Video Generation on Complex Datasets"
_ICLR.cc/2020/Conference — Reject_

### Official Review · AnonReviewer1 · 2019-10-18
**Official Blind Review #1**

**Rating:** 3

**Review:**


Summary:
This paper presents a method for training a generative adversarial network on high resolution videos and complex datasets. They propose decomposing the discriminator in Adversarial Networks into a spatial and temporal discriminator similar to previous works, however, the temporal discriminator downsamples the input using average pooling before forwarding it through the network. In experiments, the presented method outperforms previous state-of-the-art methods in the used metrics. In addition, the videos generated from the Kinetic-600 dataset in a non-conditional setting are the most realistic looking up to date.


Pros:
+ Best generated video quality out there (In the Kinetics-600 dataset)
+ Numerically outperforms baselines in all datasets

Weaknesses / comments:
* What is the message of the paper?
- It is not clear to me whether this paper is about a new method or if it’s about an empirical study of the network size for video prediction algorithms. If the paper is about a new method, then the novelty is lacking. Decomposing the discriminator into space and time discriminators was originally proposed by MoCoGAN. Therefore, the only difference I can see is that the input to the time discriminator passes through a downsampling function (average pooling). On the other hand, if the paper is about showing that a bigger video model generates better video, the paper is lacking a more in depth evaluation. It seems that the size (frame resolution and number of frames) is only done on the Kinetics-600 dataset (Table 1). However, it is not clear to me whether there is a conclusion to be made from these results about the model size. The FID seems to favor smaller models and IS favors bigger models? But the last (biggest) model does not increase the IS? I am not sure what to make of these results. In addition, there is no such results on the other datasets (UCF101 and BAIR) or on the frame conditioned experiments for Kinetics-600. Do the authors have any comments or clarifications on this?


* Lacking proof that the network is not just overfitting to each class.
- It seems that the authors train separate models for each video class. The generated videos are the best I have seen in such complicated dataset. The authors claim that the fact that the Kinetics-600 dataset is large automatically removes the concern of overfitting. However, I would like to see  evidence that this is true. Can the authors do something like sample a video and find its nearest neighbor in the training data and compare them to see if the model is simply memorizing the training set? Something like this could serve as evidence that the network is not simply memorizing the training data.


* FVD Evaluation on Robot Push (marginal improvement based on FVD paper)
- In Table 3, the authors present quantitative comparisons for frame-conditioned video prediction with state-of-the-art methods. The authors of the FVD paper state the following:

“In Figure 5 it can be seen that when the difference in FVD is smaller than 50, the agreement with human raters is close to random (but never worse).”

The difference between DVD-GAN and SAVP / Video Transformer is below 50. Does this mean that this result may not significant? It seems that basically the top 3 methods are performing similarly. Can the authors comment on this?


* Comparing 11 frame conditional generation with 16 frame unconditional generation (DVD-GAN)?
- In Table 4, the authors are comparing DVD-GAN-FP, Video Transformer and DVD-GAN. DVD-GAN-FP clearly outperforms Video Transformer. However, the authors make a comment about the DVD-GAN result being much better than the other two and that “the synthesis model’s improved performance on this task seems to indicate that the advantage of being able to select videos to generate is greater than the advantage of having a ground truth distribution of starting frames.”. However, the DVD-GAN model generates 16 frames, and not 11 like the other 2. Is this comparison correct? Also the other 2 methods (DVD-GAN-FP and Video Transformer) are constrained by what they must predict given input frames, does this fact affect the comparison / conclusion by the authors?


* Claim of diversity but no evidence in the video prediction setting (frame conditional generation).
- The authors claim that their method generates diverse videos, however, there is no palpable evidence that this is true. Something like showing examples of different videos generated given the same condition should be good or per-frame evaluations like PSNR and SSIM with error bars of the N number of samples generated given the same condition could also serve as proof that the generation is diverse.


* Video results only provided for Kinetics-600
- It looks like the authors only provide actual video files for the un-conditional generative model for Kinetics-600. Could the authors provide videos of the frame conditioned experiments as well (UCF101, BAIR and Kinetics-600)?


* Argument in paper: “The synthesis model’s improved performance on this task seems to indicate that the advantage of being able to select videos to generate is greater than the advantage of having a ground truth distribution of starting frames”
- I do not fully agree with this argument. Optimally, a frame-conditional model should identify all objects/background observed in the input frame and object/background dynamics if multiple frames are given. Given the optimally identified factors, video generation/prediction should be much better or the same as a purely generative model. The fact that the experiments in this paper show that the conditional model is not as good as the non-conditional model is most likely due to the model and/or objective function and not the generation scenario. Do the authors have any comments on this?


Conclusion:
This paper does indeed present the best video generation up-to-date given the dataset difficulty. However, the paper itself has multiple issues as stated above. Please try to address these in the next revision.


**Experience Assessment:**

I have published in this field for several years.

**Review Assessment: Checking Correctness Of Derivations And Theory:**

N/A

**Review Assessment: Checking Correctness Of Experiments:**

I carefully checked the experiments.

**Review Assessment: Thoroughness In Paper Reading:**

I read the paper thoroughly.

---

> ### Author Response · Authors · 2019-11-07
> **Author Response to Blind Review #1**
>
> Thank you for your review! We address each of your comments separately below.
>
> > ... It is not clear to me whether there is a conclusion to be made from these results about the model size...
>
> Our motivation for reporting a variety of Kinetics-600 results was to establish strong baselines for future work to improve on, not to claim that larger models are fundamentally better. We believe that Kinetics-600 represents a challenging benchmark dataset for future work, and that putting forward several variations of the dataset for different users would be the most useful. For instance, 12 frames of 64x64 video requires substantially less compute, and is therefore more generally accessible than 48 frames of 128x128.
>
> > It seems that the authors train separate models for each video class ...  The authors claim that the fact that the Kinetics-600 dataset is large automatically removes the concern of overfitting ...
>
> It is not correct that we train different models for each class -- we have now clarified this by an additional sentence in Section 4.1.1. In the class-conditional case we train a single model: the generator is told which class each generation should belong to, and both discriminators are told the class of each video. This is described in Appendices A.2/A.3.
>
> We will carry out the nearest neighbor analysis for future revision; in the meantime, we refer the reader to Appendix A of the BigGAN paper [1] which carried out such analysis for the BigGAN generative image model on ImageNet, and showed that there was no memorization in their case. Please also note that in Appendix D.2 we include samples from class and latent interpolations: another way to demonstrate that DVD-GAN does not simply memorize data from the training set, because a model which did that would be unable to generate a smooth interpolation of videos in the latent space.
>
> > ... [On < 50 score difference in FVD] Does this mean that this result may not significant? ... Can the authors comment on this? ...
>
> We have not undertaken a systematic study of FVD’s correlation with human ratings on our samples, so we did not wish to claim an equivalence of DVD-GAN-FP’s generations to those of VideoTransformer (which was developed concurrently). Uncertainty around the FVD metric for BAIR was one reason for our focus towards Kinetics: despite extremely similar FVD scores on BAIR, DVD-GAN-FP substantially outperforms VideoTransformer on FVD for Kinetics-600 prediction.
>
> > Comparing 11 frame conditional generation with 16 frame unconditional generation (DVD-GAN)?
>
> DVD-GAN-FP and VideoTransformer both generate 11 frames conditioned on an initial 5 and the FVD score is computed on the concatenation of the 5 conditioning frames and the 11 generated frames. In other words, the ground truth statistics are calculated on 16 frame clips from Kinetics, just like DVD-GAN's synthesis samples, which allows us to meaningfully compare their scores as done in Section 4.2.1.
>
>
> > Claim of diversity but no evidence in the video prediction setting (frame conditional generation)
> Our claim of diversity in the conclusion was focused on the video synthesis case, and we did not claim that our method was able to generate diverse continuations of a given frame conditioning, however we have made this more clear in the revision just posted.
>
> > Video results only provided for Kinetics-600
> We are unfortunately unable to provide frame-conditioned samples from Kinetics for licensing reasons. We have however included continuations from BAIR in Appendix B and synthesis samples from UCF-101 in Appendix D of the revision just uploaded (we did not run video prediction tests on UCF-101).
>
> > I do not fully agree with this argument. Optimally, a frame-conditional model should identify all objects/background observed in the input frame and object/background dynamics if multiple frames are given....
>
> Our intuition was the same as yours. We included the paragraph you cite as commentary on the (surprising) result that these two similar models (DVD-GAN and DVD-GAN-FP) with identical objectives -- but with different conditioning -- exhibit a different ordering (according to the FVD metric) than expected. We believe that the reason for such a behavior is that with the current learning capacity of our model it is easier for it to focus on several modes of the natural video distribution (the unconditional setting) rather than fully grasp the entire distribution and be able to continue an arbitrary video (the conditional setting).
>
> [1] https://arxiv.org/abs/1809.11096

---

### Official Review · AnonReviewer2 · 2019-10-22
**Official Blind Review #2**

**Rating:** 6

**Review:**

The paper proposes a class-conditional GAN model for video generation called DVD-GAN. The generator uses a single latent variable and uses ConvGRU modules and ResNet blocks to generate N frames. The model uses a dual discriminator, with one discriminator that discriminates individual frames, i.e. an image discriminator, and one that discriminates the whole video. This is similar to the MoCoGAN model, with the main difference being that the video discriminator operates on a smaller resolution video, thus reducing the dimensionality of the input to discriminate. The model is used to generate videos after being trained on the large-scale Kinetics-600 dataset, which contains multiple examples and has a lot of variability across videos. The main contribution of the paper is to successfully train this large GAN model on the very large-scale Kinetics dataset. The samples from the model are very visually appealing and are qualitatively  superior to any previous video prediction model.

While the paper mostly focuses on scaling up current models, it achieves significantly better qualitative results than previous models on a very challenging dataset, and therefore I believe it should be accepted as it is a significant advance in the field which probably will lead to follow-up work based on the model proposed here.

However, there are a number of things that could be improved/minor comments:

- Further details about the generator should be included in the main body of the paper, only having a figure to describe its architecture is not enough when the model and how to scale it up are key contributions.
- The authors introduce a FID score for video which is similar to FVD, but FID is only used to report results in one experiment, while FVD is used for the rest of the experiments. Since the community uses FVD and there is a publicly available implementation of this metric, I'd suggest that the authors also include FVD scores in Table 1 to help reproduce the results. This is important since the FID metric is not explained thoroughly in the paper and small implementation details matter when using these metrics.
- The related work section is missing many references to video prediction models, please add them to the paper. Some examples include:
Decomposing motion and content for natural video sequence prediction. Villegas et al. ICLR 2017
Unsupervised Learning of Disentangled Representations from Video. Denton and Birodkar. NIPS 2017
Predrnn: Recurrent neural networks for predictive learning using spatiotemporal lstms. Wang et al. NIPS 2017
PredRNN++, ContextVP, ...
- Additional metrics for the BAIR experiment, including LPIPS and SSIM. While FVD correlates well with human judgement, LPIPS does so as well and provides another evaluation of the model. Furthermore, metrics such as SSIM as used in SVG-LP can help better understand how well do the models cover the ground-truth sequence for given context frames.
- Qualitative results for the BAIR experiment. Since most current models are not trained on Kinetics, qualitative samples for the BAIR dataset would help to qualitatively compare current methods to DVD-GAN.
- In practice people have found that it is very difficult to train BigGAN-like models on images and videos. A common difficulty is that training diverges after a number of iterations, with the model starting to show mode collapse and big oscillations in terms of FID scores. Since the models are trained for a big number of iterations, have you observed these kind of issues with different hyperparameter configurations? If so, did you find any strategies to address it?

**Experience Assessment:**

I have published one or two papers in this area.

**Review Assessment: Checking Correctness Of Derivations And Theory:**

N/A

**Review Assessment: Checking Correctness Of Experiments:**

I carefully checked the experiments.

**Review Assessment: Thoroughness In Paper Reading:**

I read the paper thoroughly.

---

> ### Author Response · Authors · 2019-11-07
> **Author Response to Blind Review #2**
>
> Thanks for your review. We’ve detailed responses to your comments individually below.
>
> > Further details about the generator should be included in the main body of the paper...
> We give a detailed overview of the generator architecture in Appendix A.2 but were unable to fit all the details in the main body for length considerations. If there are important details you feel are worth moving to the main text we would be happy to do so.
>
> > ...I'd suggest that the authors also include FVD scores in Table 1 to help reproduce the results...
> The revision just uploaded makes explicit that the FID metric we propose is identical to FVD except for using a different classifier for feature extraction -- one which has been trained on Kinetics-600 instead of Kinetics-400. We also use hidden layer activations as features instead of the logits -- more in line with the FID metric used for images. The new revision details in Appendix A.4 the (extremely minor) changes needed to be made to the publicly available FVD code to represent our metric. Nevertheless, we agree with the reviewer in the value of providing both metrics and will update the paper with FVD metrics for the synthesis case in the next revision.
>
> > The related work section is missing many references to video prediction models...
> Thank you for the references! We’ve added them and other relevant references in the new revision.
>
> > Additional metrics for the BAIR experiment ...
> The paper which introduced FVD reported a substantially weaker correlation between SSIM and human judgement, which was our motivation for focusing on FVD, but nevertheless we have just run an evaluation for SSIM on our best model. We would like to include LPIPS results as well, however this will require more setup and will not be complete until a later revision.
>
> For 16 frames of BAIR with 1 conditioning frame, our per-frame SSIM scores are:
> [1.0, 0.92, 0.88, 0.87, 0.84, 0.84, 0.82, 0.82, 0.82, 0.81, 0.81, 0.80, 0.79, 0.79, 0.78, 0.78]
>
> VideoTransformer does not give explicit numbers (just a plot) so an exact numeric comparison is not possible. Judging from Figure 2a in VideoTransformer, DVD-GAN’s SSIM frame-1 score of 0.92 is on par with VideoTransformer and slightly below VideoFlow’s 0.95. Frames 2-6 of DVD-GAN are better than any other model, with the rest of the frames being roughly on par with VideoFlow and VideoTransformer. The final value of 0.78 at frame 16 is slightly lower than VideoTransformer.
>
> > Qualitative results for the BAIR experiment...
> Agreed, the revision just uploaded includes samples from BAIR in the Appendix B as well as synthesis samples from UCF-101 in Appendix D.
>
> >... difficult to train BigGAN-like models on images and videos ... any strategies to address it?
> With an extremely small amount of initial hyperparameter tuning (settling on the results listed in the paper) our model was very stable on large datasets. While DVD-GANs will eventually diverge on smaller datasets like UCF-101 and BAIR, at no point up to 1M iterations (the largest number of steps we have trained these models for) did a DVD-GAN diverge on the full Kinetics dataset, which we attribute to the dataset’s increased complexity and difficulty to be overfit.

---

> > ### Comment · AnonReviewer2 · 2019-11-13
> > **Response**
> >
> > Thanks for the response. I acknowledge that I have read it and at this point I have no further questions.

---

### Official Review · AnonReviewer3 · 2019-10-23
**Official Blind Review #3**

**Rating:** 3

**Review:**

Summary:
This paper tackles the problem of efficient video generation. The authors present a Dual-Video-Discriminator Generative Adversarial Network (DVD-GAN) composed of an image-level spatial discriminator and video-level temporal discriminator. DVD-GAN achieves state-of-the-art results when benchmarked against the FID, IS and FVD quantitative metrics. Compared to previous video generation works, DVD-GAN is the first model to present compelling qualitative results on the UCF-101 and Kinetics-600 dataset.

Motivation:
Efficient video generation using GANs remains a significant challenge as it exacerbates all the issues associated with image generation using GANs. There are also increased memory and computation costs due to the 3D nature of video and the requirement for temporal modelling.

Main Contributions:
1. The authors propose to spatially down-sample the input into the video-level temporal discriminator.
2. The authors propose to temporally sub-sample video frames for the image-level spatial discriminator

Secondary Contributions:
1. The authors set a benchmark for video generation on the Kinetics-600 dataset. This is significant due to the scale and complexity of this dataset in comparison to other datasets in the video generation literature.

----------------
Pros:
+ The paper is fairly well-written and clear
+ The paper recognizes and tackles a significant challenge in video generation when using GANs.
+ The ablation studies and experiments offer clarity with regard to the performance of the proposed model on different datasets and with different sampling strategies.
+ This paper presents compelling results for high-resolution (i.e. > 64x64) video generation on complex datasets.


Cons:
The main weakness of this paper is that it does not represent a significant advancement in our understanding of video generation using GANs. The novelty of DVD-GAN is also limited with respect to prior video GAN literature. DVD-GAN appears to be a straight-forward application of the BIG-GAN [5] family of models to video generation. Subsequent experiments and results are tailored for this particular model with little to no applicability to prior models or generalization. This is a significant weakness given that this paper's main contribution is a discriminator component for video GAN architectures.

Other Observations:
- Dual Video Discriminator GANs were introduced in MoCoGAN [1], the naming of this model may mislead readers into thinking that this is the first such discriminator architecture.
- Subsampling of the discriminator has already been explored in TGANv2 [2].

Points of Improvement:
- Given that the video discriminator is the main contribution, it should be benchmarked against previous video generator models in the literature such as TGAN [3], MoCoGAN [1] and TGANv2 [2]
- Given that the reproducibility of these results (at this moment in time) is difficult outside of a few organisations, can the authors provide a more thorough exposition of the efficiency aspects of their proposed discriminator. Currently, the best argument for the efficiency claims is the 58% reduction in pixels processed per video but what does this mean concretely, with respect to memory, computation and time efficiency?


Final notes:
- Authors should clear up the confusion with class-conditional vs unconditional UCF101 results by clearly labeling class-conditional vs unconditional results. Furthermore, they should provide a detailed section on how the UCF101 metrics were achieved, it is known that video GAN metrics are highly sensitive to even the decision of when to normalize inputs to the evaluation network (see Appendix B of [4])

- As impressive as the results are on Kinetics-600, they ultimately are dampened if they cannot be reproduced or built upon. The original Kinetics-600 dataset (based on youtube clips) is no longer available as youtube has either blocked, geo-blocked or removed a significant portion of this dataset. In light of this, do the authors plan to provide access to this dataset or should we place less significance on the Kinetics results?

-------

Current Review Decision:
Although this paper does present state-of-the-art results for all video generation datasets and carries out thorough experimentation to demonstrate this. The DVD-GAN model rides significantly on the backbone of the BIG-GAN model to achieve this. The Dual Video Discriminator by itself does not provide a novel contribution for a conference such as ICLR as it has already been proposed [1] and discriminator subsampling has been explored in different forms previously [2]. Crucially, the claims on efficiency are not sufficiently explored and it is unclear what they are and how they would translate to other video GAN models in the literature. As such, I lean towards rejecting this paper in its current form.


----------------
[1] - https://arxiv.org/abs/1707.04993
[2] - https://arxiv.org/abs/1811.09245
[3] - https://arxiv.org/abs/1611.06624
[4] - https://arxiv.org/abs/1909.12400
[5] - https://arxiv.org/abs/1809.11096


**Experience Assessment:**

I have published one or two papers in this area.

**Review Assessment: Checking Correctness Of Derivations And Theory:**

N/A

**Review Assessment: Checking Correctness Of Experiments:**

I carefully checked the experiments.

**Review Assessment: Thoroughness In Paper Reading:**

I read the paper thoroughly.

---

> ### Author Response · Authors · 2019-11-07
> **Author Response to Blind Review #3**
>
> Thank you for the review. We’ve responded to your comments below (including some additions included in the revision just posted).
>
> > ... can the authors provide a more thorough exposition of the efficiency aspects of their proposed discriminator.
>
> DVD-GAN as described required 7.77 GB of memory per accelerator and ran with a step time of 968.8 ms. An identical model without downsampling in the temporal discriminator or subselection in the spatial discriminator (akin to MoCoGAN [1]) required 8.47 GB of memory per accelerator and ran with a step time of 1,415.9 ms: meaning the DVD-GAN decomposition improved memory usage by 9% and wall clock time by a substantial 46.1%.
>
> > Given that the video discriminator is the main contribution, it should be benchmarked against previous video generator models in the literature such as TGAN [3], MoCoGAN [1] and TGANv2 [2]
>
> The goal of the UCF-101 experiments in Section 4 was precisely to give quantitative comparison to these models. In 4.3 we point out that we expect less subsampling (as in TGAN or MoCoGAN) to be better but more expensive, and show an experiment verifying it, and in Section 3.2 we discuss the differences between our subsampling and that of TGANv2 [2]. We also show that DVD-GAN is still able to outperform TGAN, MoCoGAN, and TGANv2 by 15.5, 15 and 3 Inception Score respectively.
>
> > Authors should clear up the confusion with class-conditional vs unconditional UCF101 results ... they should provide a detailed section on how the UCF101 metrics were achieved...
>
> The version of the paper just uploaded as revision contains a refactored UCF-101 results section. There, we report the IS results both without class conditioning (27.38 vs 24.34 of TGANv2), and with class conditioning (32.97 vs 15.83 of Conditional TGAN). In both settings, our results set the state of the art. The revision also contains more details of the IS computation for UCF-101, which are unchanged from TGAN’s evaluation. The paper you reference does highlight the importance of details of evaluation, but like them we use the publicly available TGAN evaluation code (what they call “Algorithm A”), and so our numbers are comparable.
>
> > ...should we place less significance on the Kinetics results?
>
> Kinetics is one of the largest class-conditional video datasets publicly released that has a high, human-vetted level of quality. We are not aware of any systematic change in the dataset’s availability which prevents its use, with many papers being published on it as an action classification benchmark (and a newer version -- which we do not use in this paper -- being released just a few months ago). If there are accessibility problems with the dataset, we suggest reaching out to the dataset creators, but we do not think it should be factored into the significance of our (or any other paper’s) results on this dataset, since the Kinetics dataset is not a part of our contribution.
>
> [1] https://arxiv.org/abs/1707.04993
> [2] https://arxiv.org/abs/1811.09245

---

### Public Comment · ~Girolamo_Cardano1 · 2019-09-26
**unconditional vs class conditional UCF101 results**

Hi,

1. UCF101 results

If i understood the paper correctly, all results presented in this paper are class-conditional video generation results.
In that case it should be made explicitly clear in Table 2, for the UCF-101 results. The authors currently present their class-conditional results against unconditional results from previous video generation models.

The distinction between class-conditional and unconditional video generation is crucial because *class conditioning significantly boosts performance for video generation models*. Presenting class conditional results together with unconditional video generation results will mislead nonexpert readers, as to the performance of this model on this dataset.

For example; in the original TGAN paper (Table 4 of [1]), the inception score for unconditional video generation on the UCF101 dataset is 11.85±.07.
However, Table 4 in the TGAN paper also shows that their class-conditional model achieves an inception score of 15.83±.18.
One can observe from just this result, that the performance of the conditional TGAN model is still better than results achieved by every other subsquent unconditional video generation model with the exception of TGANv2. TGANv2 currently holds the state-of-the-art result for unconditional video generation.

I highly advise that in relation to the UCF101 dataset, the authors should present results for their model on unconditional video generation. This will allow for an accurate comparison against prior work. It is currently unclear to me whether DVD-GAN is a better video generation model than prior work such as TGANv2.

Failing that, I advise that the authors consider including their results in a seperate table for conditional video generation, where they can compare against other conditional results from models such as TGAN. The conditional TGAN results are currently not mentioned anywhere in this paper.

It would also aid clarity if the authors can explicitly mention how they calculated the IS and FID for the UCF101 dataset (Appendix B.1 in the DVD-GAN paper). TGAN and TGANv2 calculate the inception score in slightly different ways but these differences do in reality impact the final score (especially for the FID metric).

TGAN method - "sampling 10,000 times from the latent random variable, and derived a rough standard deviation by repeating this procedure four times"

TGANv2 [2] method - "2,048 samples were used for computing IS and FID. The standard deviation was calculated by performing the same procedure 10 times"



2. Questions about the main contributions

Given the name of the model presented in this paper, it implies that one of the main contributions of this paper is the Dual-Video Discriminator.  This is further emphasised the more one reads through the paper.
Dual video discriminators have already been presented in previous models such as MoCoGAN [3]. This is acknowledged by the authors in section 3.1. The main difference between the discriminator in this model and that of MoCoGAN being that DVD-GAN's video discriminator operates on a downsampled version of the input?

Interestingly, the DVD-GAN generator appears to be one of the most interesting contributions. MoCoGAN uses a GRU on the latents and TGANv2 uses a convolutional LSTM on it's latents. If i understand the paper correctly, DVD-GAN has a generator with recurrence at every spatial resolution except the last?  If so, it would be good to provide a thorough exposition of the generator and explore how recurrence at different spatial resolutions affects performance.

----
[1] -  https://arxiv.org/abs/1611.06624
[2] - https://arxiv.org/abs/1811.09245
[3] - https://arxiv.org/abs/1707.04993

---

> ### Author Response · Authors · 2019-10-16
> **Response to Anonymous Comment**
>
> > If i understood the paper correctly, all results presented in this paper are class-conditional video generation results.
> This is a very good point, and we’re thankful to the commenter for bringing this to our attention as it was a sincere oversight. Class-conditioning information does aid the performance of our network. Since you’ve posted this comment, we’ve fixed the problem and retrained DVD-GANs on UCF-101 without any class information. The model is unchanged from the description in the paper, though some hyperparameters were adjusted for the unconditional setting. In particular, the learning rate is set to 5e-5 and 6e-5 for G and D respectively, and the channel multiplier is 128 for G and 64 for D. DVD-GAN without any class conditioning achieves an inception score of 27.38 +- 0.53 on UCF-101, still improving over TGANv2’s 24.34 +- 0.15 in the same setting.  We also found that data augmentation as used in TGANv2 was crucial to prevent early overfitting and collapse in the unconditional setting. We will describe this experiment in the next version of the paper. Thank you again for noticing this discrepancy!
>
> We also calculated FID statistics on UCF-101. TGANv2 does not describe how they calculate the features for their FID metric, so we assumed they used the logits of C3D as their features as in [1]. DVD-GAN gets 956 FID in this metric, substantially better than TGANv2’s 3620 (though we cannot be sure this is the same metric as TGANv2 does not say which features it uses).
>
> > It would also aid clarity if the authors can explicitly mention how they calculated the IS and FID for the UCF101 dataset
> We calculated Inception Score identically to TGANv2 (2048 samples done 10 times to calculate standard deviation), and we will add a sentence in revision clarifying this.
>
> > The main difference between the discriminator in this model and that of MoCoGAN being that DVD-GAN's video discriminator operates on a downsampled version of the input?
> Yes, as described in sections 3.1 and 3.2 this seemingly minor change significantly changes the role of the per frame discriminator and makes it essential for good performance.
>
> > If so, it would be good to provide a thorough exposition of the generator and explore how recurrence at different spatial resolutions affects performance.
> We felt that our discriminator decomposition was a more important design choice to focus on. We do acknowledge the usefulness of ablating the RNNs in the generator though, and will add that in a future version.
>
>
> [1] https://arxiv.org/abs/1812.01717

---

### Decision · Program_Chairs · 2019-12-19

**Decision:**

Reject

**Comment:**

This paper addresses the tasks of video generation and prediction and shows impressive results on the datasets such as Kinetics-600. There is a reviewer disagreement on this paper. AC can confirm that all three reviewers have read the rebuttal and have contributed to a long discussion. The reviewers have raised the following concerns that were viewed as critical issues when making the final decision: R1 and R3 expressed the concerns regarding limited technical novelty of the proposed approach in light of the prior works, e.g. MoCoGAN and TGANv2. R3 suggests, that the proposed method shows advantage that might be due to the large computational resources available to train the model. Providing a comparison of the proposed model and the relevant baselines on the Kinetics dataset is desirable to access the benefits of the proposed approach (R1).
AC also agrees with the R2 about the potential impact this work could have in the community. However, given that the reviewers have raised important concerns and have given suggestions, the paper needs too many revisions for acceptance at this time. We hope the reviews are useful for improving and revising the paper.